# The Efficacy of Medical Interventions for Free-Floating Thrombus in Cerebrovascular Events: A Systematic Review

**DOI:** 10.3390/brainsci14080801

**Published:** 2024-08-09

**Authors:** Fairoz Jayyusi, Majd M. AlBarakat, Habib H. Al-Rousan, Mohmmad M. Alawajneh, Abdel Rahman Alkasabrah, Mo’tasem Abujaber, Mohammed E. Aldabbas, Mustafa Abuelsamen, Yahya Alshgerat, Yahia Sayuri, Nazeeh Alhertani, Mohammad BaniAmer, Issa Shari, James Robert Brašić

**Affiliations:** 1Faculty of Medicine, Jordan University of Science and Technology, Irbid 22110, Jordan; fwjayyusi18@med.just.edu.jo (F.J.); mmalbarakat20@med.just.edu.jo (M.M.A.); hhalrousan19@med.just.edu.jo (H.H.A.-R.); mmalawajneh183@med.just.edu.jo (M.M.A.); ayalkasabrah180@med.just.edu.jo (A.R.A.); mat07170776@ju.edu.jo (M.A.); mdabbas2000@hotmail.com (M.E.A.); maabuelsamen19@med.just.edu.jo (M.A.); yfalshuqirat182@med.just.edu.jo (Y.A.);yahya.sayouri12@hotmail.com (Y.S.); nwalhertani18@med.just.edu.jo (N.A.); mibaniamer21@med.just.edu.jo (M.B.).; iishari181@med.just.edu.jo (I.S.); 2Department of Psychiatry, New York City Health and Hospitals/Bellevue, New York, NY 10016, USA; 3Department of Psychiatry, New York University Grossman School of Medicine, New York University Langone Health, New York, NY 10016, USA; 4Section of High-Resolution Brain Positron Emission Tomography Imaging, Division of Nuclear Medicine and Molecular Imaging, The Russell H. Morgan Department of Radiology and Radiological Science, The Johns Hopkins University School of Medicine, Baltimore, MD 21287, USA

**Keywords:** anticoagulants, antiplatelets, cardioembolism, central nervous system infarction, confidence interval, odds ratio, prognosis, statins, stroke, transient ischemic attack (TIA)

## Abstract

Although free-floating thrombus (FFT) poses a significant risk of stroke or transient ischemic attack (TIA), optimal management strategies are uncertain. To determine the state-of-the-art of medical interventions for FFT, we conducted a systematic review of the efficacy of various medical interventions and factors influencing FFT resolution and recurrence. A comprehensive search of Embase, PubMed, and ScienceDirect identified 61 studies encompassing 179 patients with FFT-related stroke or TIA treated with anticoagulants, antiplatelets, or their combinations. Primary outcomes assessed were stroke recurrence and thrombus resolution. Statistical analyses (Fisher’s exact test, chi-square test, Mann–Whitney test, and Kruskal–Wallis test) utilized significance set at *p* < 0.05. Over a median follow-up of 7 months, thrombus resolution occurred in 65% of patients, while 11.2% experienced recurrence, primarily as TIAs. Cardioembolism was significantly less common in resolved cases (*p* = 0.025). Combination therapy (antiplatelets, anticoagulants, and statins) significantly enhanced clot resolution (OR 11.4; 95% CI 1.436–91.91; *p* = 0.021) compared to monotherapies. Ulcerated plaque was a significant predictor of recurrence (OR 8.2; 95% CI 1.02–66.07; *p* = 0.048). These findings underscore the superiority of combination therapy in FFT management and highlight the need for targeted interventions in patients with ulcerated plaques to mitigate recurrence risk.

## 1. Introduction

The American Heart Association defines stroke or central nervous system (CNS) infarction as brain, spinal cord, or retinal cell death attributable to ischemia, based on CNS focal ischemic injury evidenced by objective evidence such as imaging in a defined vascular distribution or clinical evidence with symptoms persisting ≥24 h or until death [1]. Ischemic strokes can be traced back to two main causes: thrombotic and embolic [2]. Understanding the causes and locations of ischemic strokes is important for determining the outcome and prognosis. Thrombotic events develop as clots locally within the blood vessels and are commonly stagnant. However, free-floating thrombus (FFT), an uncommon variant, has recently gained attention due to advancements in minimally invasive and high-resolution imaging modalities [3].

Free-floating thrombus (FFT), also termed mobile or intraluminal thrombus, is a blood clot that is attached to the arterial wall with blood flow circulating its distal component [4]. It was first documented in the 1960s, yet limited research has been conducted on it due to the similarities it has with other conditions such as embolic thrombus and mobile plaques, which have made reaching a widely accepted definition challenging [3]. Current studies show that atherosclerosis is the most common cause of FFT, followed by dissections, malignancies, and trauma [5]. The prevalence of FFT among acute stroke patients is estimated to be between 1.6% and 3.2% [6]. Additionally, men are twice as likely to develop FFT compared to women, with the average age of diagnosis being 57.6 [4]. It is diagnosed through several imaging modalities, including magnetic resonance angiography (MRA), computed tomography angiography (CTA), digital subtraction angiography (DSA), and duplex ultrasound (DUS) [3,7].

FFT is a significant cause of recurrent ischemic stroke and transient ischemic attack (TIA) when not managed properly [8]. Treatment approaches for FFT include pharmacological therapy, surgical therapy, or a combination of both. However, the existing literature shows no consensus on which approach is optimal [3]. This limited knowledge, along with the absence of agreed-upon guidelines, poses a challenge for neurologists in managing FFT. Even when medical therapy is chosen, it is unclear whether to use anticoagulants alone or with antiplatelets or single or dual antiplatelet agents [4,7].

Since a meta-analysis is appropriate for this clinical problem, we constructed a systematic review and meta-analysis including the extant publications. Although our meta-analysis [9] concluded that surgical interventions produced better outcomes than medical interventions, this was not consistent with prior studies [3,4,10]. When we presented our recent systematic review and meta-analysis at the 2024 Annual Meeting of the American Academy of Neurology [9], our colleagues pointed out that the quality of the studies included was too poor to draw firm conclusions on the superiority of medical or surgical therapies. Therefore, we realized that a meta-analysis of medical and surgical therapies may not be the optimal approach to assess the state of the art at this time. To determine the optimal strategy for FFT resulting in minimal morbidity and mortality, we sought to compare therapeutic approaches for FFT and the corresponding outcomes by means of a systematic review of the current literature for medical interventions. We also sought to identify sociodemographic characteristics associated with favorable outcomes for FFT. We sought to develop guidelines for providers to utilize precision medicine to tailor comprehensive treatment plans for individuals with FFT.

## 2. Materials and Methods

### 2.1. Study Design and Search Protocol

A review of the literature was conducted in accordance with the Preferred Reporting Items for Systematic Reviews and Meta-Analyses (PRISMA) guidelines (Figure 1) [11]. This study was not registered for PRISMA. Two investigators independently searched the following online databases: Embase, PubMed, and ScienceDirect. We based our search strategy (Table 1) on the keywords “thrombus” and “stroke”, with synonyms. Additionally, the reference sections of the relevant literature were manually inspected to identify unique records. The literature review was conducted from 1 January 1990 to 1 March 2024.

### 2.2. Inclusion/Exclusion Criteria

Inclusion criteria were as follows: (1) patients diagnosed with FFT by MRA, CTA, DSA, or DUS, leading to either stroke or TIA; (2) patients treated with anticoagulation therapy alone or in combination with antiplatelet therapy, dual antiplatelet therapy (DAPT), or mono antiplatelet therapy; (3) primary reported outcomes, such as stroke recurrence or assessment of thrombus status post-medical intervention; (4) patients aged 18 years or older; (5) reports in English, French, or Spanish.

FFT was identified on imaging by either a contrast filling defect (visible on angiography, CTA, or MRA) or a hyperechoic formation without a flow signal (detected on DUS) originating from the proximal carotid wall and extending freely into the distal arterial lumen [3]. We adopted this practical imaging definition to address the clinical management challenge when FFT is strongly suspected. Consequently, observed movement during the cardiac cycle (seen only on angiography or DUS), partial or complete dissolution on follow-up imaging, or pathological confirmation of FFT was not required for our diagnostic criteria [12,13].

Secondary studies such as literature reviews, systematic reviews, and meta-analyses were excluded. Studies involving patients with total vessel occlusion were excluded. Patients diagnosed with FFT solely during peri-procedural periods were also excluded to mitigate potential biases arising from procedural influences on diagnosis. Furthermore, studies relying mainly on the pathologic diagnosis of FFT or lacking clinical correlation were not included.

### 2.3. Quality Assessment

Two investigators independently evaluated the quality, including the risk of bias, for each study. Discrepancies between the reviewers were resolved by a third investigator. We employed the Methodological Index for Non-Randomized Studies (MINORS) for the risk of bias assessment [14]. Each criterion was scored as follows: 0 (not present), 1 (reported but inadequate), or 2 (reported and adequate). Criteria include a clearly stated aim, consecutive patient inclusion, prospective data collection, endpoints relevant to the study aim, unbiased endpoint assessment, appropriate follow-up duration, follow-up loss under 5%, and prospective study size calculation.

For studies with a comparative group like Schwartzmann et al. [15], additional items were considered: adequate control group, contemporary groups, baseline group equivalence, and adequate statistical analysis. Scoring for single-arm studies rated 8 or lower indicates a high risk of bias, 8 to 12 denotes medium risk, and 12 or higher indicates low risk. For comparative studies, a total score exceeding 17 indicates a low risk of bias, while less than 17 signifies a high risk.

Additionally, we utilized the Joanna Briggs Institute (JBI) critical appraisal tools for case reports and case series [16]. Items were assessed as “yes”, “no”, “not clear”, or “not applicable”. Studies were classified as high risk of bias if fewer than half of the items were rated “yes”, moderate risk if 50–70% were “yes”, and low risk if over 70% were “yes”.

### 2.4. Data Extraction

Four reviewers (M.Abuel., Y.A., Y.S., and N.A) independently used a well-designed online extraction form to extract the upcoming data. The first part included the summary characteristics of the included studies (name of first author, year of publication, and study design). The second part included the baseline information of the participants (sample size, age, gender, follow-up period, risk factors, type of event, diagnosis, follow-up imaging methods, vessel involved, and medical intervention). Finally, the third part included outcome data. Endnote software (version 21) [17] was utilized to detect duplicates, organize retrieved records, and coordinate screening of the records. The data extraction process was carried out by two reviewers (M.M.Ala. and A.R.A.). Any disagreements that arose were resolved through discussion and agreement with a senior author.

### 2.5. Data Synthesis and Analysis

Continuous variables are presented as either mean ± standard deviation (SD) or median [interquartile range], depending on their adherence to a normal distribution. Categorical variables are reported as number (percentage) values. Statistical comparison of categorical variables was conducted using Fisher’s exact test or the chi-square test, while nonparametric data were analyzed using the Mann–Whitney test or the Kruskal–Wallis test. A significance threshold of *p* < 0.05 was applied for statistical significance. All analyses were carried out using R software (version 3.4.0, R Foundation, Vienna, Austria) [18].

## 3. Results

### 3.1. Literature Search and Methodological Quality

We identified 61 studies [8,12,13,15,19,20,21,22,23,24,25,26,27,28,29,30,31,32,33,34,35,36,37,38,39,40,41,42,43,44,45,46,47,48,49,50,51,52,53,54,55,56,57,58,59,60,61,62,63,64,65,66,67,68,69,70,71,72,73,74,75,76] to be included according to our inclusion/exclusion criteria. Included studies reported FFT in 179 patients who were medically treated. The types of studies included were case reports (47), case series (9), retrospective cohorts (3), and prospective cohorts (2) (Table 2).

All studies assessed by the Methodological Index for Non-Randomized Studies (MINORS) criteria [14] were considered either low risk of bias or moderate risk of bias except Schwartzmann et al. [15] which was considered to have a high risk of bias. For the included case reports and case series, all had either a low or moderate risk of bias except for Papadoulas et al. [70] which had a high risk of bias according to the Joanna Briggs Institute (JBI) critical appraisal tools [16].

### 3.2. Study Characteristics

The average age of the participants was 58.7 years, with a standard deviation of 15.4 years. The median follow-up period for the entire cohort was 7 months, ranging from 0.3 to 115 months. There was a male predominance of 63.7%. Medical treatment was divided into four groups: 62/179 (34.6%) received anticoagulants with antiplatelets, 28/179 (15.6%) received antiplatelets alone, 75/179 (41.9%) received anticoagulants alone, and 18/179 (10.1%) received anticoagulants with antiplatelets and a high dose of statins.

### 3.3. Demographic and Medical Characteristics

Among the entire sample, 117 (65%) individuals exhibited partial or complete resolution of clots, while 62 (35%) patients experienced non-resolution or progression of the clots.

The analysis revealed no significant differences between patients with resolved clots and those with non-resolved clots relative to various factors, including age, gender, atherosclerosis, smoking, and other risk factors as indicated in Table 3. However, a notable distinction emerged in the follow-up duration, with a median of 6 months (range: 0.25–115) for the resolved group and a median of 8.5 months (range: 0.03–108) for the non-resolved group (*p*-value = 0.034).

Cardioembolism, as a source and etiology of clots, also presented a substantial difference between the two groups. All of the 9 patients with this etiology had a resolved clot, and they represented 7.7% of all resolved clots. This difference between groups was statistically significant with a *p*-value of 0.025.

Data on arterial involvement were available for 179 individuals. Among these, 149 patients (83.2%) manifested lesions in the extracranial circulation, while 30 patients (16.8%) exhibited lesions in the intracranial circulation. Among the 117 patients with resolved FFT, 91 patients (77.8%) presented with extracranial thrombosis, whereas the remaining 25 patients (21.4%) demonstrated intracranial thrombosis. Data is missing for one of the patients with resolved FFT. In contrast, among the 62 patients without FFT resolution, 55 patients (88.7%) displayed extracranial thrombosis, while 7 patients (11.3%) had intracranial thrombosis. Although no statistically significant difference was observed between the two groups (*p* = 0.089), these results indicate discernible patterns in thrombus resolution across both intra and extracranial circulations. 

### 3.4. Type of Medical Therapy and Thrombus Outcome

Medications employed in stroke management were classified into four categories as illustrated in Table 4. The solitary use of either anticoagulant therapy or antiplatelet therapy did not demonstrate any significant difference when compared with the combined use of antiplatelet and anticoagulant therapies (*p* = 1.201, *p* = 0.780), respectively. Conversely, the utilization of a combination therapy involving antiplatelet agents, anticoagulants, and statins showed a higher probability of clot resolution, with an odds ratio of 11.4 and a confidence interval (CI) of 1.436–91.91, along with a *p*-value of 0.021.

### 3.5. Factors Affecting Recurrence 

Within our cohort, 20 patients (11.2%) experienced a recurrence; 3 (15%) experienced a stroke and 17 (85%), a TIA. Among the 20 patients, 14 (70%) were male, and the mean age was 61.6 ± 13.7. Atherosclerosis, identified as the most prevalent risk factor within this group, was present in eight of these patients. Individuals exhibiting ulcerated plaque faced an elevated risk of recurrence, indicated by an odds ratio (OR) of 8.2, along with a confidence interval (CI) of 1.02–66.07 and a *p*-value of 0.048 (see Table 5).

## 4. Discussion

To determine the optimal strategy for FFT resulting in minimal morbidity and mortality, we sought to compare therapeutic approaches for FFT and the corresponding outcomes by means of a systematic review of the current literature. We also sought to identify sociodemographic characteristics associated with favorable outcomes for FFT. We sought to generate recommendations for providers to facilitate interventions for FFT without adverse effects.

Similar to prior systematic reviews and meta-analyses [3,4,9,10,77], our systematic review is hindered by the poor quality of the data. Direct comparison between our review and prior reviews and meta-analyses [3,4,9,10,77] is feasible for those parameters included in prior studies and our study. In other words, while other studies [3,4,9,10] considered both medical and surgical interventions for FFT, we restricted our review to medical interventions. Therefore, comparisons with prior studies [3,4,9,10] would be like comparing apples with oranges. Since there are so many differences among studies [3,4,9,10,77], comparison of the different studies is beyond the scope of this review about medical interventions for FFT.

We compared the four medical management strategies for FFT (anticoagulants, antiplatelets, anticoagulants + antiplatelets, and anticoagulants + antiplatelets + statins) (Table 4). The only therapy that significantly improved clot resolution was the combined treatment of antiplatelets, anticoagulants, and statins. This suggests that statins play an imperative role when combined with dual or monotherapy, increasing their effectiveness. This finding aligns with the findings of Ní Chróinín et al. [10] which found that statin administration to patients at the onset of ischemic stroke improved outcomes significantly. This is attributable to the atherosclerotic nature of thrombi formation [78]. Our study underscores the importance of statins’ addition to antithrombotic management of stroke, not only in its prevention.

Additionally, there was no significant difference in clot resolution between anticoagulants or antiplatelets compared to combination therapy. This further suggests that, when used independently, they have comparable efficacies. Nevertheless, a case for dual therapy can be made, as it addresses multiple aspects of thrombus formation and stabilization and is evidenced in the literature to decrease the risk for recurrent ischemic stroke [8].

While our study primarily focused on FFT medical management efficacy, we have had consequential results that further consolidate our understanding of the disease and factors influencing clot resolution and cerebrovascular accident (CVA) recurrence.

In our study, all patients with FFT caused by cardioembolism attained full resolution. Thus, patients with FFT caused by cardioembolism may benefit greatly from medical interventions to attain clot resolution. The value of accurately identifying the etiology of the FFT is further heightened in our results, which encourage more research to be done with a bigger sample size on treatment efficacy in FFT caused by cardioembolism.

Our results indicated no significant differences in age, gender, atherosclerosis, smoking, or other common risk factors between patients with resolved and unresolved clots. Patients with non-resolved clots were monitored longer and thus had a higher chance of detecting any CVA recurrences than patients with resolved clots.

Similar to the literature, atherosclerosis was the most prevalent risk factor within our sample. Our results have shown minor differences in association with recurrence between patients with studied risk factors. However, only one risk factor illustrated a significance in recurrence rates. Our analysis showed that patients with ulcerated plaques experienced higher rates of recurrences than those without.

Generally, the results indicate that the primary focus should be on specific thrombus characteristics and the effectiveness of initial treatment strategies. Nonetheless, offering more aggressive or tailored interventions to ulcerated plaque patients for secondary prevention might prove to be especially useful.

Although a comprehensive discussion of other interventions for FFT in cerebrovascular events is beyond the scope of this article about medical interventions for FFT, we would like to point out that several promising endovascular and surgical tools have benefited some patients. Carotid endarterectomy has benefited some patients with an ischemic cerebrovascular event due to FFT. In particular, the 2023 European Society for Vascular Surgery (ESVS) carotid guidelines include therapeutic anticoagulation, but not intravenous thrombolysis, for symptomatic patients with evidence of FFT, and surgical or endovascular thrombectomy for symptomatic patients with FFT who develop recurrent symptoms on anticoagulation [79]. The beneficial effects of carotid endarterectomy have been confirmed by a small case series [80].

### Limitations

A key challenge in evaluating FFT and interpreting our results is the inconsistency in data reporting, including variations in follow-up durations and the lack of a standardized definition of FFT. Additionally, small sample sizes further complicate the analysis. Despite our rigorous approach, the quality of the included studies, mostly case reports and case series, limits our findings. Poor quality of data was a major problem of our systematic review, like prior systematic reviews [3,4,9,10,77]. Because we had mostly case reports and case series, a comprehensive quality assessment was not possible. This made it difficult to compare medical interventions for recurrence and clot resolution, introducing uncertainty. Moreover, there were insufficient data to determine if dual antiplatelet therapy is better than a single agent, or to establish the optimal duration of therapy. However, this review lays the groundwork for future well-designed studies.

## 5. Conclusions

This research highlights the complexities of managing free-floating thrombus (FFT) in cerebrovascular events such as ischemic strokes and transient ischemic attacks. Our study shows potential for the pharmacological management of FFT through combined antithrombotic and lipid-lowering therapies. We recommend that physicians include statins alongside anticoagulants and antiplatelets to optimize thrombus resolution and reduce the recurrence of cerebrovascular events. Additionally, treatment and follow-up strategies should be tailored to the etiology and characteristics of the thrombus.

We urge future researchers to explore FFT further, as it is an under-studied type of thrombus. More clinical trials and prospective studies with larger sample sizes on FFT management protocols are needed. Increased research efforts should focus on understanding FFTs, especially those caused by cardioembolism and those associated with ulcerated plaques.

Reducing the burden of ischemic strokes caused by FFT can be achieved by improving patient outcomes and developing standardized guidelines based on rigorous evidence.

## Figures and Tables

**Figure 1 brainsci-14-00801-f001:**
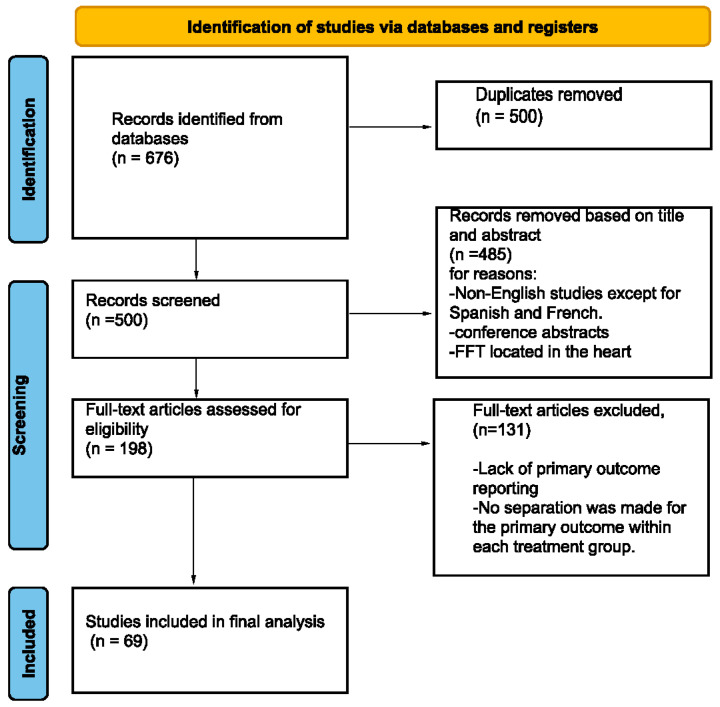
Preferred Reporting Items for Systematic Reviews and Meta-Analyses (PRISMA) [11] flowchart of the screening process.

**Table 1 brainsci-14-00801-t001:** Search terms in the databases for a systematic review of free-floating thrombus (FFT).

Database	Search Terms
PubMed	(“arteries”[MeSH Terms] OR “arteries”[All Fields]) AND (“thrombus”[MeSH Terms] OR “thrombus”[All Fields]) AND (“stroke”[MeSH Terms] OR “stroke”[All Fields] OR “TIA”[All Fields] OR “transient ischemic attack”[MeSH Terms]) AND (“medical treatment”[All Fields] OR “therapy”[MeSH Terms] OR “management”[All Fields])
EMBASE	(‘arteries’/exp OR ‘arteries’) AND (‘thrombus’/exp OR ‘thrombus’) AND (‘stroke’/exp OR ‘stroke’ OR ‘TIA’ OR ‘transient ischemic attack’/exp) AND (‘medical treatment’/exp OR ‘therapy’ OR ‘management’)
ScienceDirect	(“arteries” AND “thrombus” AND (“stroke” OR “TIA” OR “transient ischemic attack”) AND (“medical treatment” OR “therapy” OR “management”))

**Table 2 brainsci-14-00801-t002:** Characteristics of studies in a systematic review of free-floating thrombus (FFT).

First Author/Year	Study Type	Number of Patients	MINORS	Total Risk—JBI
Schwartzmann 2023 [15]	Prospective Cohort	1	16	
Amarenco 1992 [58]	Case series	1		Moderate
Vassileva 2015 [54]	Prospective Cohort	5	11	
Jaberi 2014 [53]	Retrospective Cohort	14	15	
Gülcu 2014 [8]	Case Series	18	11	
Vellimana 2013 [75]	Retrospective Cohort	13	11	
Chua 2012 [12]	Retrospective Cohort	25	11	
Mokin 2012 [40]	Case Series	18		Moderate
Delgado 2011 [65]	Case Series	3		Low
Pagni 2011 [39]	Case Series	3		Low
Choi 2009 [52]	Case Series	9	12	
Bouly 2005 [30]	Case Series	3	10	
Choukroun 2002 [19]	Case Series	3		Low
Laperche 1997 [55]	Case Series	3		Low
Adam 2023 [73]	Case Report	1		Low
Zedde 2023 [26]	Case Report	1		Moderate
Ferraù 2022 [43]	Case Report	2		Moderate
Gotfrit 2022 [31]	Case Report	1		Low
Chiang 2022 [23]	Case Report	1		Low
Shiozaki 2021 [68]	Case Report	1		Moderate
Papadoulas 2021 [70]	Case Report	1		Low
Christou 2021 [50]	Case Report	1		Low
Koneru 2021 [49]	Case Report	1		Moderate
Kim 2021 [48]	Case Report	1		Low
Prasad 2021 [47]	Case Report	1		Low
Hsieh 2021 [20]	Case Report	1		Moderate
Gomez-Arbelaez 2020 [44]	Case Report	1		Low
Viguier 2020 [32]	Case Report	1		Low
Atsina 2020 [27]	Case Report	1		Moderate
Lalla 2020 [25]	Case Report	1		Low
Hosseini 2020 [22]	Case Report	1		Low
Ikenouchi 2019 [34]	Case Report	1		Moderate
Akpinar 2019 [71]	Case Report	1		Low
Fitzpatrick 2018 [57]	Case Report	1		Low
Roy 2017 [69]	Case Report	1		Low
Stein 2017 [62]	Case Report	3		Moderate
Oh 2017 [33]	Case Report	1		Low
Karapurkar 2016 [46]	Case Report	1		Low
Omoto 2016 [35]	Case Report	1		Low
Frias Vargas 2016 [29]	Case Report	1		Moderate
Karapanayiotides 2015 [59]	Case Report	1		Low
Renard 2014 [38]	Case Report	1		Low
Batur Caglayan 2013 [67]	Case Report	1		Low
Delgado 2013 [64]	Case Report	1		Moderate
Elijovich 2013 [74]	Case Report	2		Low
Bok 2013 [72]	Case Report	1		Low
Park 2013 [24]	Case Report	1		Low
Morelli 2012 [56]	Case Report	1		Low
Alurkar 2012 [51]	Case Report	4		Moderate
Jaberi 2012 [53]	Case Report	3		Moderate
Switzer 2011 [63]	Case Report	1		Moderate
Tran 2011 [36]	Case Report	1		Moderate
Nakajima 2008 [45]	Case Report	1		Low
Yamagami 2005 [6]	Case Report	1		Low
Karapanayiotides 2004 [41]	Case Report	2		Low
Mathew 2002 [76]	Case Report	1		Moderate
Sogawa 2001 [28]	Case Report	1		Low
Masuo 2000 [44]	Case Report	1		Low
Ko 1997 [60]	Case Report	2		Low
Akins 1996 [21]	Case Report	3		Moderate
Buscaglia 1993 [61]	Case Report	1		Low

Abbreviations: JBI: Joanna Briggs Institute [16]; MINORS: Methodological Index for Non-Randomized Studies [14].

**Table 3 brainsci-14-00801-t003:** Predictors of resolution of free-floating thrombus (FFT).

	Total	Resolved (117)	Not Resolved (62)	*p*-Value
Age (years)	58.7 (15.4)	58.9 (14.7)	58.6 (16.7)	0.89
Follow-up (months)	60.5 (20–102)	6 (0.25–115)	8.5 (0.03–108)	0.034
Gender (male)	114 (63.7%)	74 (63.2%)	40 (64.5%)	0.867
Atherosclerosis	52 (29.1%)	34 (29%)	18 (29%)	0.997
Ulcerated plaque	7 (3.9%)	4 (3.4%)	3 (4.8%)	0.641
Ruptured plaque	9 (5%)	8 (6.8%)	1 (1.6%)	0.128
Atrial fibrillation	8 (4.5%)	4 (3.4%)	4 (6.5%)	0.35
Valvular heart disease	1 (0.6%)	0	1 (1.6%)	0.168
Hypercoagulable states	33 (18.4%)	22 (18.8%)	11(17.7%)	0.862
Dissection	6 (3.4%)	3 (2.6%)	3 (4.8%)	0.421
Cryptogenic stroke	3 (1.7%)	3 (2.6%)	0	0.204
Cardioembolism	9 (5%)	9 (7.7%)	0	0.025
Current smoker	25 (13.9%)	17 (14.5%)	8 (12.9%)	0.765
Diabetes mellitus	25 (14%)	14 (12%)	11(17.7%)	0.289
Ischemic heart disease	15 (8.4%)	9 (7.7%)	6 (9.7%)	0.648
Heart failure	1 (0.6%)	1 (0.9%)	0	0.465
Dyslipidemia	50 (27.9%)	33 (28.2%)	17 (27.4%)	0.911
Hypertension	57 (31.8%)	36 (30.8%)	21 (33.9%)	0.672
Alcohol use	3 (1.7%)	3 (2.6%)	0	0.204
Previous stroke/TIA	11 (6.1%)	8 (6.8%)	3 (4.8%)	0.596

Abbreviation: TIA: Transient ischemic attack.

**Table 4 brainsci-14-00801-t004:** Thrombus outcomes of medical therapies.

Medical Therapy	OR	CI 95%	*p*-Value
Antiplatelets + anticoagulants	Reference	Reference	Reference
Anticoagulants	1.201	0.601–2.40	0.604
Antiplatelets	0.78	0.317–1.92	0.588
Antiplatelet + anticoagulants + statins	11.486	1.436–91.91	0.021

Abbreviations: CI: Confidence interval; OR: Odds ratio.

**Table 5 brainsci-14-00801-t005:** Predictors of stroke/TIA recurrence.

	Odds Ratio	95% Confidence Interval		*p*-Value
		Lower	Upper	
Male	0.939	0.503	1.75	0.842
Atherosclerosis	0.865	0.2175	3.439	0.837
Ulcerated plaque	8.211	1.0205	66.072	0.048
Ruptured plaque	1.495	0.1883	11.862	0.704
Atrial fibrillation	2.074	0.1828	23.539	0.556
Hypercoagulable states	0.219	0.0269	1.783	0.156
Dissection	1.496	0.1242	18.018	0.751
Cardioembolism	0.551	0.0551	5.515	0.612
Current smoker	0.667	0.1379	3.227	0.615
Diabetes mellitus	0.281	0.0259	3.048	0.297
Ischemic heart disease	0.974	0.0962	9.857	0.982
Dyslipidemia	0.697	0.146	3.332	0.652
Hypertension	0.878	0.216	3.569	0.856
Previous stroke/TIA	1.037	0.0953	11.276	0.976

Abbreviation: TIA: Transient ischemic attack.

## Data Availability

All data are included in the text.

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
