# Peer review of "The Efficacy of Medical Interventions for Free-Floating Thrombus in Cerebrovascular Events: A Systematic Review"

_brainsci, 2024, doi:10.3390/brainsci14080801_

Round 1

Reviewer 1 Report

Comments and Suggestions for Authors

The authors report a systematic review on medical treatment of floating thrombus (FT) They identified 61 studies, with overall  179 patients, treated with anticoagulants, antiplatelets, statins or their combinations.  With  a follow-up of 7 months, thrombus resolution occurred in 65% of patients, Combination therapy (antiplatelets, anticoagulants, and statins) better enhanced clot resolution compared to monotherapies.

The topic is interesting and the design is well described. It would be interesting, in the discussion to spend some words on surgery of thrombus, even if it is not the goal of the study.

Reviewer 2 Report

Comments and Suggestions for Authors

Dear Authors, 

congratulate for the paper which has, to my opinion touched a very important issue in stroke secondary prevention. It presents the current standing in antithrombotic treatment of free-floating thrombus, a rare, but high risk condition where guidelines are missing. It also emphasizes the importance of therapeutic approach, as well as use of statins. My opinion is that the topic is of high interest, methods appropriate, and that the paper helps in reaching clinical decisions.

Reviewer 3 Report

Comments and Suggestions for Authors

This is a nice systematic review that tackles an important issue with high clinical relevance. However, some comments can improve the quality of the paper. 

Literature search is limited and some major indexing engine were not included. Please explain your rationale for choosing these sites. 

PRIMSA chart is the old version. Please use the new one

A comprehensive quality assessment of the included studies was not performed. Giving that most studies are case report, you can add this to the discussion or limitations. 

Please describe in detail the data extraction process. 

Author Response

Point 1: This is a nice systematic review that tackles an important issue with high clinical relevance. However, some comments can improve the quality of the paper. 

Response 1: We thank Reviewer 3 for the perceptive comments. We have addressed the issues raised by Reviewer 3 to improve the methods and results as requesed.

Point 2: Literature search is limited and some major indexing engine were not included. Please explain your rationale for choosing these sites. 

Response 2: We chose the indexing engines based on the practices of other systematic reviews for free-floating thrombus in cerebrovascular events. We utilized the example of the search engines of recent systematic reviews cited in our manuscript as follows:

(3)    Fridman, S.; Lownie, S. P.; Mandzia, J. Diagnosis and management of carotid free-floating thrombus: A systematic literature review. Int. J. Stroke 2019, 14, 247–256 DOI: 10.1177/1747493019828554.

(4)   Bhatti, A. F.; Leon, L. R.; Labropoulos, N.; Rubinas, T. L.; Rodriguez, H.; Kalman, P. G.; Schneck, M.; Psalms, S. B.; Biller, J. Free-floating thrombus of the carotid artery: literature review and case reports. J. Vasc. Surg. 2007, 45, 199–205 DOI: 10.1016/j.jvs.2006.09.057.

Point 3: PRIMSA chart is the old version. Please use the new one

Response 3: We have added the current version for the PRISMA chart as Figure 1 as follows:

Point 4: A comprehensive quality assessment of the included studies was not performed. Giving that most studies are case report, you can add this to the discussion or limitations. 

Response 4: Because we included case reports (47), case series (9), retrospective cohorts (3), and prospective cohorts (2), we improvised a quality assessment as follows:

2.3. Quality Assessment

Two investigators independently evaluated the quality including the risk of bias for each study. Discrepancies between the reviewers are resolved by a third investigator. We employed the Methodological Index for Non-Randomized Studies (MINORS) for the risk of bias assessment 12. Each criterion was scored as follows: 0 (not present), 1 (reported but inadequate), or 2 (reported and adequate). Criteria include a clearly stated aim, consecutive patient inclusion, prospective data collection, endpoints relevant to the study aim, unbiased endpoint assessment, appropriate follow-up duration, follow-up loss under 5%, and prospective study size calculation.

For studies with a comparative group like Schwartzmann et al. 13, additional items were considered: adequate control group, contemporary groups, baseline group equivalence, and adequate statistical analysis. Scoring for single-arm studies rated 8 or lower indicates a high risk of bias, 8 to 12 denotes medium risk, and 12 or higher indicates low risk. For comparative studies, a total score exceeding 17 indicates a low risk of bias, while less than 17 signifies a high risk.

Additionally, we utilized the Joanna Briggs Institute (JBI) critical appraisal tools for case reports 14 and case series 14. Items were assessed as “yes”, “no”, “not clear”, or “not applicable”. Studies were classified as high risk of bias if fewer than half of the items were rated “yes”, moderate risk if 50-70% were “yes”, and low risk if over 70% were “yes”.

We acknowledged the absence of a comprehensive quality assessment in the limitations as follows:

4.1. Limitations

A key challenge in evaluating FFT and interpreting our results is the inconsistency in data reporting, including variations in follow-up durations and the lack of a standardized definition of FFT. Additionally, small sample sizes further complicate the analysis. Despite our rigorous approach, the quality of the included studies, mostly case reports and case series, limits our findings. Poor quality of data was a major problem of our systematic review, like prior systematic reviews 3,4,9,75 . Because we had mostly case reports and case series, a comprehensive quality assessment was not possible. This made it difficult to compare medical interventions for recurrence and clot resolution, introducing uncertainty. Moreover, there was insufficient data to determine if dual antiplatelet therapy is better than a single agent or to establish the optimal duration of therapy. However, this review lays the groundwork for future well-designed studies.

Point 5: Please describe in detail the data extraction process. 

Response 5: We have provided a section about the data extraction process as follows:

2.4. Data Extraction

Four reviewers (M.Abuel., Y.A., Y.S. and N.A) independently used a well-designed online extraction form to extract the upcoming data. The first part included the summary characteristics of the included studies (name of first author, year of publication and study design). The second part included the baseline information of the participants (sample size, age, gender, follow-up period, risk factors, type of event, diagnosis, follow-up imaging methods, vessel involved, medical intervention. Finally, the third part included outcome data. The data extraction process was carried out by two reviewers (M.M.Ala. and A.R.A). Any disagreements that arose were resolved through discussion and agreement with a senior author.

Reviewer 4 Report

Comments and Suggestions for Authors

I am grateful to the editor for the opportunity to review the manuscript by Fairoz Jayyusi et al. "The Efficacy of Medical Interventions for Free-Floating Thrombus in Cerebrovascular Events: A Systematic Review". This article is devoted to the evaluation of drug treatment for Free-Floating Thrombus in Cerebrovascular Events. There are few such observations in general, so the authors of the article used a rare design - they combined in their article an individual analysis of articles with descriptions of clinical cases (both individual and series) and small cohort studies. In total, 61 studies with consideration of 179 patients were included in the analysis. This approach allowed the authors to form 4 groups of drug therapy and compare them in terms of effectiveness - in terms of thrombus resolution and prevention of recurrent cerebrovascular events. To some extent, these results may be interesting. While reviewing, I had the following comments and questions:

1. The design of this study is more suitable for an individual meta-analysis, since it combines individual observations into a single database.

2. When discussing the results obtained, they should be compared with the data of previously conducted reviews and meta-analyses (for example, with the results of the study by Fridman et al.). Also, an analysis of previous publications in response to these reviews (for example, Castillo-Torres SA et al.) should be included in the Discussion.

3. I also missed in the Discussion consideration of a wider range of interventions for Free-Floating Thrombus in Cerebrovascular Events, including endovascular and surgical ones.

References:

1. Fridman S, Lownie SP, Mandzia J. Diagnosis and management of carotid free-floating thrombus: A systematic literature review. Int J Stroke. 2019 Apr;14(3):247-256. doi: 10.1177/1747493019828554.

2. Castillo-Torres SA, Soto-Rincón CA, Góngora-Rivera F. Readers' response to: "Diagnosis and management of carotid free-floating thrombus: A systematic literature review". Int J Stroke. 2019 Dec;14(9):NP3-NP4. doi: 10.1177/1747493019886248.

Comments on the Quality of English Language

No comments

Author Response

Point 1: I am grateful to the editor for the opportunity to review the manuscript by Fairoz Jayyusi et al. "The Efficacy of Medical Interventions for Free-Floating Thrombus in Cerebrovascular Events: A Systematic Review". This article is devoted to the evaluation of drug treatment for Free-Floating Thrombus in Cerebrovascular Events. There are few such observations in general, so the authors of the article used a rare design - they combined in their article an individual analysis of articles with descriptions of clinical cases (both individual and series) and small cohort studies. In total, 61 studies with consideration of 179 patients were included in the analysis. This approach allowed the authors to form 4 groups of drug therapy and compare them in terms of effectiveness - in terms of thrombus resolution and prevention of recurrent cerebrovascular events. To some extent, these results may be interesting. While reviewing, I had the following comments and questions:

Response 1: We thank Reviewer 4 for the perceptive comments. We have addressed the issues raised by Reviewer 4 to improve the methods and results as requested.

Point 2:. The design of this study is more suitable for an individual meta-analysis, since it combines individual observations into a single database.

Response 2: We agree that the design of the study is more suitable for an individual meta-analysis. Therefore, we conducted a systematic review and meta-analysis to present at the recent annual meeting of the American Academy of Neurology. The feedback from the presentation convinced us that the quality of the studies in the meta-analysis was too poor to generate meaningful results. Therefore, we proceeded to conduct a systematic review with rigid criteria to generate important findings that merit publication now so that colleagues around the world can benefit from our conclusions.

            To explain these experiences, we added statements in the last paragraph of the introduction and the associated citation as follows:

Since a meta-analysis is appropriate for this clinical problem, we constructed a systematic review and meta-analysis including the extant publications. Although our meta-analysis 9 concluded that surgical interventions produced better outcomes than medical interventions, this was not consistent with prior studies 3,4,10. When we presented our recent systematic review and meta-analysis at the 2024 Annual Meeting of the American Academy of Neurology 9, our colleagues pointed out that the quality of the studies included was too poor to draw firm conclusions the superiority of medical or surgical therapies. Therefore, we realized that a meta-analysis of medical and surgical therapies may not be the optimal approach to assess the state of the art at this time. To determine the optimal strategy for FFT resulting in minimal morbidity and mortality, we sought to compare therapeutic approaches for FFT and the corresponding outcomes by means of a systematic review of the current literature for medical interventions. We also sought to identify sociodemographic characteristics associated with favorable outcomes for FFT. We sought to develop guidelines for providers to utilize precision medicine to tailor comprehensive treatment plans for individuals with FFT.

. . .

(9)        Jayyusi, F.; Jabeiti, S.M.; Taha, M.J.J.; Badir, S.; Alawajneh, M.M.; Momani, A.; Hasan, A.; Abulesamen, M., Shawawrah, M. Comparative efficacy of procedural vs. medical interventions in the management of free-floating thrombus: a systematic review and meta-analysis. Neurology 2024, 102 (17 supplement 1) DOI: 10.1212/WNL.0000000000204822.

Point 3: When discussing the results obtained, they should be compared with the data of previously conducted reviews and meta-analyses (for example, with the results of the study by Fridman et al.). Also, an analysis of previous publications in response to these reviews (for example, Castillo-Torres SA et al.) should be included in the Discussion.

References:

  1. Fridman S, Lownie SP, Mandzia J. Diagnosis and management of carotid free-floating thrombus: A systematic literature review. Int J Stroke. 2019 Apr;14(3):247-256. doi: 10.1177/1747493019828554.
  2. Castillo-Torres SA, Soto-Rincón CA, Góngora-Rivera F. Readers' response to: "Diagnosis and management of carotid free-floating thrombus: A systematic literature review". Int J Stroke. 2019 Dec;14(9):NP3-NP4. doi: 10.1177/1747493019886248.

Response 3: We thank Reviewer 4 for this thoughtful comment. We agree that previously conducted reviews and meta-analyses and the responses the these reviews merit inclusion in our article. We have include both the publications suggested by Reviewer 4. The prior reviews and meta-analyses used different parameters from our manuscript. In particular, we restricted our review to a systematic review of medical interventions only utilizing strict inclusion and exclusion criteria. Prior studies have not yielded unequivocal conclusions about interventions. Therefore, we are not attempting to compare and contrast our study of medical interventions only with other studies including different parameters. We addressed this point by a paragraph in the Discussion as follows:

Similar to prior systematic reviews and meta-analyses 3,4,9,10,76, our systematic review is hindered by the poor quality of the data. Direct comparison between our review and prior reviews and meta-analyses 3,4,9,10,76, is feasible for those parameters included in prior studies and our study. In other words, while other studies 3,4,9,10 considered both medical and surgical interventions for FFT, we restricted our review to medical interventions. Therefore, comparisons with prior studies 3,4,9,10 would be like comparing apples with oranges. Since there are so many differences among studies 3,4,9,10,76, comparisons of the different studies is beyond the scope of this review about medical interventions for FFT.

Point 4: I also missed in the Discussion consideration of a wider range of interventions for Free-Floating Thrombus in Cerebrovascular Events, including endovascular and surgical ones.

Response 4: We thank Reviewer 4 for the perceptive comments. We have addressed the issues raised by Reviewer 3 by including in the Discussion consideration of a wider range of interventions for Free-Floating Thrombus in Cerebrovascular Events, including endovascular and surgical ones as follows:

Although a comprehensive discussion of other interventions for FFT in cerebrovascular events is beyond the scope of this article about medical interventions for FFT, we would like to point out that several promising endovascular and surgical tools have benefited some patients. Carotid endarterectomy has benefited some patients with an ischemic cerebrovascular event due to FFT In particular, the 2023 European Society for Vascular Surgery (ESVS) carotid guidelines include therapeutic anticoagulation, but not intravenous thrombolysis for symptomatic patients with evidence of FFT, and surgical or endovascular thrombectomy for symptomatic patients with FFT who develop recurrent symptoms on anticoagulation 79. The beneficial effects of carotid endarterectomy have been confirmed by a small case series 80.

. . .

(79)    AbuRahma, A. F. The New 2023 European Society for Vascular Surgery (ESVS) Carotid Guidelines: the transatlantic perspective. Eur. J. Vasc. Endovasc. Surg. 2023, 65, 5-6 [editorial] DOI: 10.1016/j.ejvs.2022.04.011

(80)       Aldridge, B.; Coleman, D.; Wilson, S.; Solis, M. Results of carotid endarterectomy for free-floating thrombus of the carotid artery. Ann. Vasc. Surg. 2021, 77, P337 [abstract] https://doi.org/10.1016/j.avsg.2021.10.020

Round 2

Reviewer 4 Report

Comments and Suggestions for Authors

The authors have corrected the text of the manuscript and responded to my comments. I have no other comments.

Comments on the Quality of English Language

No comments